# Evidence of reassortment of avian influenza A (H2) viruses in Brazilian shorebirds

Luciano M. Thomazelli[1], João Renato Rebello Pinho[2], Erick G. Dorlass[2], Tatiana Ometto[1], Carla Meneguin[1], Danielle Paludo[3], Rodolfo Teixeira Frias[4], Patricia Luciano Mancini[4], Cairo Monteiro[1], Sophie Marie Aicher[5], David Walker[6], Guilherme P. Scagion[1], Scott Krauss[6], Thomas Fabrizio[6], Maria Virgínia Petry[7], Angelo L. Scherer[7], Janete Scherer[7], Patricia P. Serafini[8,9], Isaac S. Neto[8], Deyvid Emanuel Amgarten[2], Fernanda de Mello Malta[2], Ana Laura Boechat Borges[2], Robert G. Webster[6], Richard J. Webby[6], Edison L. Durigon[1,10], Jansen de Araujo[1]*

1 Laboratório de Pesquisa em vírus Emergentes and Laboratório de Virologia Clínica e Molecular at Biomedical Science Institute (ICB-II), University of São Paulo, São Paulo, Brazil, 2 Hospital Israelita Albert Einstein, São Paulo, São Paulo, Brazil, 3 Instituto Chico Mendes de Conservação da Biodiversidade (ICMBio), Núcleo de Gestão Integrada em Florianópolis, Santa Catarina, Brazil, 4 Instituto de Biodiversidade e Sustentabilidade (NUPEM/UFRJ), Macaé, Rio de Janeiro, Brazil, 5 Institut Pasteur, Université de Paris Cité, CNRS UMR 3569, Virus sensing and signaling Unit, Paris, France, 6 Department of Infectious Diseases, St. Jude Children's Research Hospital, Memphis, Tennessee, United States of America, 7 Laboratório de Ornitologia e Animais Marinhos, Universidade do Vale do Rio do Sinos, Rio Grande do Sul, Brazil, 8 Centro Nacional de Pesquisa e Conservação das Aves Silvestres (CEMAVE/ICMBio/MMA), Brazil, Florianópolis, Brazil, 9 Universidade Federal de Santa Catarina (UFSC), Florianópolis, Brazil, 10 Scientific Platform Pasteur-USP (SPPU), São Paulo, Brazil

* jansentequila@usp.br

**Data Availability Statement:** All relevant data are within the paper and its Supporting information files. All sequences are available on the Genbank platform (ON720806-ON720813).

## Abstract

Influenza A viruses of the H2 subtype represent a zoonotic and pandemic threat to humans due to a lack of widespread specific immunity. Although A(H2) viruses that circulate in wild bird reservoirs are distinct from the 1957 pandemic A(H2N2) viruses, there is concern that they could impact animal and public health. There is limited information on AIVs in Latin America, and next to nothing about H2 subtypes in Brazil. In the present study, we report the occurrence and genomic sequences of two influenza A viruses isolated from wild-caught white-rumped sandpipers (*Calidris fuscicollis*). One virus, identified as A(H2N1), was isolated from a bird captured in Restinga de Jurubatiba National Park (PNRJ, Rio de Janeiro), while the other, identified as A(H2N2), was isolated from a bird captured in Lagoa do Peixe National Park (PNLP, Rio Grande do Sul). DNA sequencing and phylogenetic analysis of the obtained sequences revealed that each virus belonged to distinct subtypes. Furthermore, the phylogenetic analysis indicated that the genomic sequence of the A(H2N1) virus isolated from PNRJ was most closely related to other A(H2N1) viruses isolated from North American birds. On the other hand, the A(H2N2) virus genome recovered from the PNLP-captured bird exhibited a more diverse origin, with some sequences closely related to viruses from Iceland and North America, and others showing similarity to virus sequences recovered from birds in South America. Viral genes of diverse origins were identified in one of the viruses, indicating local reassortment. This suggests that the extreme South of Brazil may serve as an environment conducive to reassortment between avian influenza virus

**Funding:** This work was supported by the Fundação de Amparo à Pesquisa do Estado de São Paulo, Grant/Award Number: 2011/13821-7 and 2017/01125-2; US National Institute of Allergy and Infectious Disease Centers of Excellence for Influenza Research and Surveillance (CEIRS) and by the American Lebanese Syrian Associated Charities (ALSAC) program, Grant/Award Number: (HHSN266200700005C); Wildlife Conservation Society (WCS) Project:, Grant/ Award Number: 2008005_WCS_OWOH and 2009005_WCS_OWOH; Fundação de Amparo à Pesquisa do Estado do Rio Grande do Sul, Grant/ Award Number: 09/0574-7. Field campaigns on PNRJ were supported by ICMBio through the GEF Mar Project for monitoring shorebirds. The funders had no role in study design, data collection and analysis, decision to publish, or preparation of the manuscript.

**Competing interests:** The authors have declared that no competing interests exist.

lineages from North and South America, potentially contributing to an increase in overall viral diversity.

## Introduction

In addition to sporadically infecting humans, causing mild-to-fatal disease, avian influenza viruses (AIVs) have contributed to multiple pandemics [1, 2]. One notable example is the A (H2N2) influenza virus, which entered the human population in 1957 through reassortment events involving avian A(H2N2) and human A(H1N1) viruses, leading to a pandemic with devastating global consequences that resulted in millions of deaths [3]. The A(H2N2) viruses ceased circulating in the human population in 1968 when a reassortment event with an avian A(H3) virus gave rise to the A(H3N2) pandemic virus [4]. Although the A(H2N2) virus disappeared from humans, A(H2) viruses still circulate in avian species. While these avian viruses are distinct from the 1957 pandemic virus, there is concern that A(H2) AIVs could reemerge due to contact between humans and domestic animal species with wild birds [3], especially in a time when the advancement of agro-agriculture has increasingly exerted pressure on green areas. Avian A(H2) viruses in wild birds typically exhibit no pathogenicity, but they can cause symptoms like sneezing, coughing, and nasal discharge in poultry and sinusitis in waterfowl [5]. Numerous subtypes of avian A(H2) viruses, including H2N1, H2N3, H2N5, H2N7, H2N8, and H2N9, have been reported in various animal hosts [6].

Our understanding of the global ecology of AIVs is limited by historically low levels of viral surveillance in South America. The current spread of highly pathogenic A(H5N1) viruses through South and Central America may provide impetus to change this with detections of the virus in birds and mammals having been reported in Argentina, Bolivia, Chile, Colombia, Costa Rica, Cuba, Ecuador, Guatemala, Honduras, Mexico, Panama, Peru, Uruguay and Venezuela [7].

Like most AIV subtypes, H2 is phylogenetically divided into two broad clades, based on geographic circulation. Unlike most AIV subtypes, however, "North American" clade sequences can be found in both classic North America and Eurasian viral genomes as a result of a trans-hemispheric transmission event and subsequent proliferation. Evidence of intercontinental spread within these lineages underscores the complexity of AIV dynamics [8].

Over the past 15 years, we have conducted influenza surveillance in wild birds at various sites in Brazil, including shorebirds at Lagoa do Peixe National Park (PNLP) and the Amazon region. Previous studies have described the genetic diversity and distribution of A(H2N2), A (H6N1), A(H9N2) and A(H12N5) AIVs in the south of Brazil [9], a lack of detection of viruses in southern grasslands, the northeastern and Pantanal wetlands [10–12], and identified A (H11N9) viruses in migratory shorebirds wintering in the Amazon [13]. Much remains to be learned about influenza in this region of the world, particularly with the spread of A(H5N1) viruses.

Some Nearctic migratory species, breed in the Canadian Arctic from May to June and migrate to Argentine and Chilean Patagonia between January and April [14]. During its southbound migration, birds enter Brazil via the northeast coast or the mouth of the Amazon River, possibly using areas in the Pantanal Biome [15]. They arrive on the southern coast of Brazil between November and January, where they congregate with various species such as gulls, terns, and shorebirds. During this time, they engage in direct contact, refuel, and subsequently continue their migration southward to Patagonia. Given their extensive migratory routes and

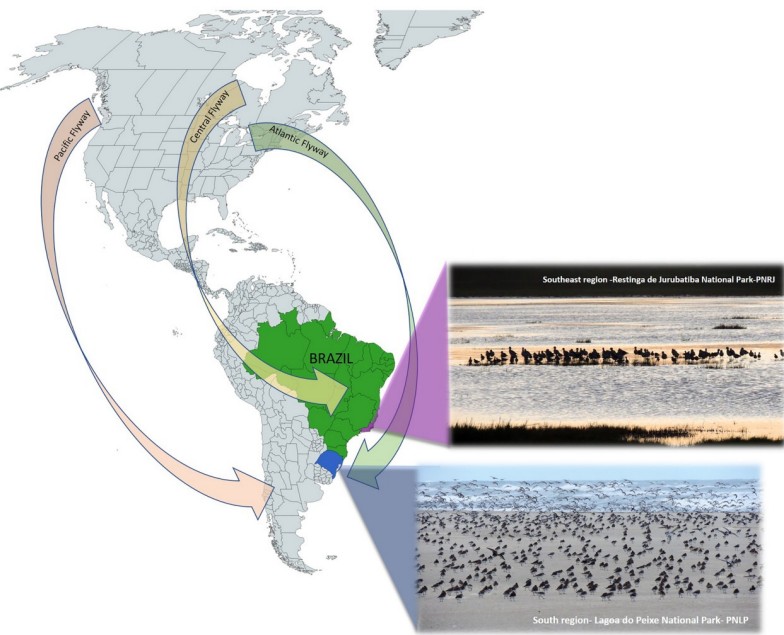

**Fig 1. Main avian migratory routes are described, with emphasis on the Atlantic route that passes through the collection sites marked on the map.** Lagoa do Peixe National Park (PNLP) in Rio Grande do Sul and Restinga de Jurubatiba National Park (PNRJ) in Rio de Janeiro. Republished from MapChart under a CC BY license, with permission from Minas Giannekas, original copyright CC BY-SA 4.0 Legal Code, 2024.

susceptibility to viruses, these birds play a significant role as transcontinental vectors of avian viruses (see Fig 1).

In this study, two A(H2) viruses were isolated from wild-caught white-rumped sandpipers (*Calidris fuscicollis*). One was an A(H2N1) virus obtained from a bird caught in Restinga de Jurubatiba National Park (PNRJ, Rio de Janeiro); the other, an A(H2N2) virus, was isolated from a bird caught in Lagoa do Peixe National Park (PNLP, Rio Grande do Sul) in the extreme south of Brazil. The viruses were characterized by genomic sequencing and compared to sequences available in GenBank. These viruses provided a glimpse into the genetic diversity of A(H2) AIVs circulating in Brazil.

## Materials and methods

### Sample collection

Combined cloacal/oropharyngeal swabs samples were collected from wild birds (mainly from *Charadriiformes*) including long and intermediate migratory waterfowl and local resident shorebirds at Restinga de Jurubatiba Natinal Park (Macaé, RJ [Lat. -22.220792 Long. -41.516596]), Rio de Janeiro, Brazil in April and October 2019. Samples collected from wild birds at an estuary in the Lagoa do Peixe National Park (PNLP) in southern Brazil [Lat. -31.257299 Long. -50.970221] that were collected in October 2012 were also included (Fig 1). Oropharyngeal/cloacal swabs were obtained from wild birds captured in mist nets and released after sampling. The swabs were placed in vials containing VTM transport medium (VTM: PBS + 10% glycerol + Antibiotics/fungizone) and immediately placed in liquid nitrogen following collection. The collection methodologies employed were as previously reported [13]. Permission was obtained for environmental sampling at the protected sandbank site at Macaé, RJ by

the Institute of Biodiversity and Sustainability NUPEM/UFRJ and National Center of Research and Conservation of Wild Birds (CEMAVE-ICMBio/MMA), (SISBIO numbers: 17565–1, 22976–8, 23159–3, 24381–3, 65230–1, 42418–9 and 14966–1).

## AIV detection

RNA was extracted from collected samples by using a MagMax TM-96 RNA Isolation Kit (Ambion, Austin, TX, USA) in accordance with the manufacturer's instructions and screened for the presence of the AIV matrix gene by one-step real-time reverse transcriptase (RT)-PCR using an AIV-M TaqMan Kit (Applied Biosystems, Foster City, CA, USA) as previously described [16].

## Virus isolation

Virus isolation was attempted from influenza positive avian samples by 10-day-old embryonated chicken eggs using the World Health Organization protocol [17]. Allantoic fluid was tested for presence of virus by hemagglutination assay and real-time RT-PCR. All procedures were conducted in a BSL3+ laboratory.

## Viral sequencing

Total RNA from positive allantoic fluids was isolated and concentrated by using an RNA Clean & Concentrator kit (Zymo Research, Irvine, USA) including a DNAse I treatment. Human ribosomal RNA was depleted with a QIAseq Fast Select RNA Removal kit (QIAGEN, Hilden, Germany). Finally, samples were submitted to random amplification following the methodology described in Greninger et al. (2015) [18]. The libraries for sequencing were prepared with a Nextera XT Kit (Illumina, San Diego, USA) using the PCR product as input, according to manufacturer's instructions. The resulting libraries were quantified, mixed at equimolar amounts and submitted to 150 bp paired-end reads using an Illumina NextSeq 550 sequencing system (Illumina). Data generated were submitted to the Varsmetagen online platform [19], where bioinformatic analyses and interpretation were carried out. The Varsmetagen virome pipeline (https://varsomics.com/varsmetagen/) performed the following steps: raw sequence metrics and filtering of low quality reads (sequences with length lower than 50 bp and Phrep score lower than 20); host decontamination by mapping reads to a close host genome (Gallus gallus assembly: GCF_016699485.2) using BWA with default parameters; first round of pathogen identification using Kraken2 [20] with a custom database including Virus, Bacteria and Human on NCBI database and complete viral genomes available in Genbank with a custom 33-kmer length; short reads assembly with Spades (version 3.13) [21]; second round of identification with contigs; search for distant homologous sequences using Hidden Markov Models (HMM) of viral proteins implemented in the eegNOG database [22] and finding confirmation through mapping and coverage metrics.

## Phylogenetic analysis

We performed a phylogenetic analysis of the obtained IAV sequences to investigate their genetic relationship to other influenza virus sequences available in GenBank. Sequence alignments were performed using MAFFT [23]. Maximum Likelihood analysis was conducted using IQ-TREE [24] for all 8 entire nucleotide AIV segments of each virus. The substitution model was chosen with the Model Finder parameter for each dataset. The statistical significance of phylogenetic groupings was tested with bootstrap analysis using 1000 replicates. Tree editing was performed with iTOL [25, 26].

The tree was drawn to scale, with branch lengths measured in the number of substitutions per site. Codon positions included were 1st+2nd+3rd+Noncoding. Alignment was checked with Geneious Prime software, and all positions containing gaps and missing data were eliminated.

## Results

### Sampling

A total of 1212 oropharyngeal/cloacal samples were collected from wild birds at Lagoa do Peixe National Park (PNLP) during 2012, belonging to different families including Ardeidae, Charadriidae, Haematopodidae, Recurvirostridae, Laridae, Rostratulidae, Tyrannidae, Furnariidae and Scolopacidae. The predominant family was Scolopacidae, with the majority being *C. fuscicolis*, accounting for 370 samples (30%). We aimed to capture distinct populations, and no animals were resampled throughout the entire collection period. In this study, we identified only one *C. fuscicollis* carrying the H2N2 subtype, representing a prevalence of 0.3% (1/370) in extreme South of Brazil. At the second site, Restinga de Jurubatiba National Park (PNRJ), sampling was conducted in 2019, resulting in a total of 118 samples representing families Ardeidae, Charadriidae, Tyrannidae, Rostratulidae, Caprimulgidae, Anatidae, Motacillidae, Jacanidaea and Scolopacidae with the majority also belonging to the same species, forty-eight *C. fuscicollis* (representing 40%). We identified an H2N1 subtype in one of these *C. fuscicollis* with a prevalence of almost 2% (1/48).

### Detection

All samples were tested for avian influenza virus RT-PCR. Two samples, one from PNLP and one from PNRJ, were positive for the H2 subtype, both from free-ranging white-rumped sandpipers (*C. fuscicollis*).

Positive samples were confirmed as influenza by sequencing of the matrix gene. The positive PNLP sample, PNLP1193, was submitted to the Department of Infectious Diseases, St. Jude Children's Research Hospital (Memphis, TN, USA), but no virus was recovered. The positive sample from PNRJ, PNRJ049, yielded a positive culture at Clinical and Molecular Virology Laboratory at Biomedical Science Institute, University of São Paulo. Both samples underwent NGS sequencing, the sequence for RJNP049 was generated from the isolated virus and PNLP1193 was generated directly from the original sample. Full length sequences of all 8 gene segments were generated (ON720798-ON720813). The samples from PNLP and PNRJ were found to contain A(H2N2) and A(H2N1) viruses, respectively.

### Phylogenetic analysis

Phylogenetic analysis of the A(H2N1) virus recovered from the PNRJ capture bird showed that all of its internal genes clustered not with gene sequences isolated from birds sampled in South America, but with sequences of influenza viruses isolated from birds sampled in North America (Figs 2 and 3, S1–S6 Figs). Analysis of the deduced HA sequence showed a cleavage site consistent with a low pathogenic AIV (VPQIESRGLF) with no variation among viruses of the H2 subtype. The neuraminidase segment was more closely related to North American than South America viruses. The most similarity related NA gene was from influenza viruses isolated from Northern Shovelers (*Spatula clypeata*) in California (99.04%) (MK995843.1) (Fig 3).

Phylogenetic analysis of the A(H2N2) virus obtained from PNLP showed that its HA clustered with multiple viruses isolated from birds sampled in Iceland and North America (Fig 2).

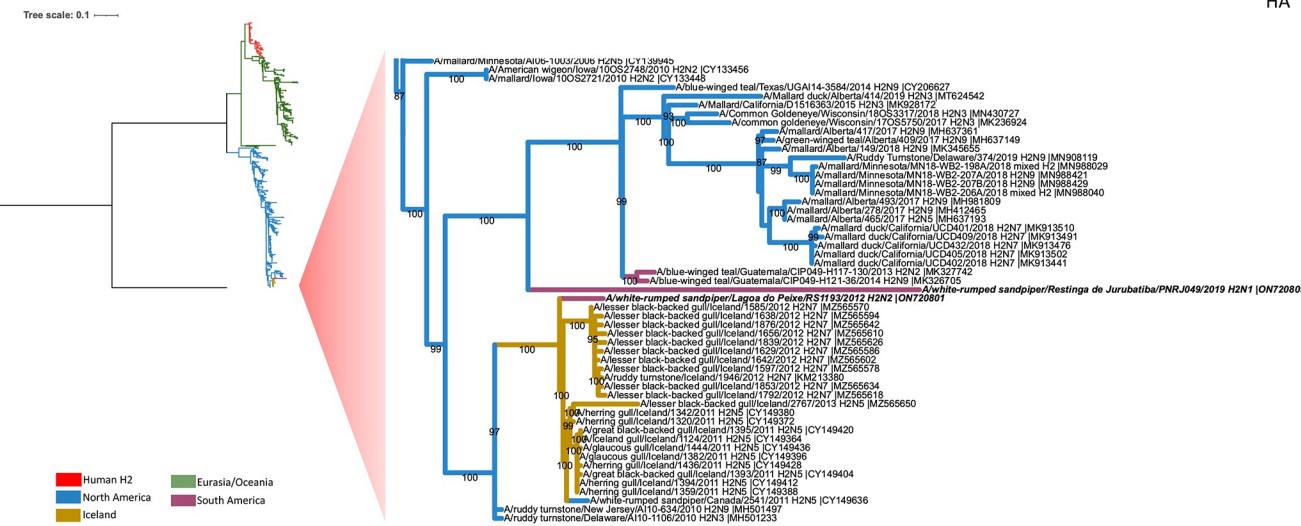

**Fig 2. Phylogenetic analysis of the PNLP and PNRJ influenza isolates hemagglutinin (HA) gene.** Brazilian samples in this study are written in a bold font. The clusters of high similarly sequences related are indicated by branches colored red (Human HA2, green (Eurasia and Oceania), blue (North America), gold (Iceland) and purple (South America). A total of 780 complete HA2 sequences available in NCBI Influenza Virus Database were used for this phylogenetic analysis. The tree was constructed using the GTR+F+I substitution model as selected by IQ-TREE Model Finder in a 1777 nt length alignment. The scale bar represents the number of substitutions per site. Bootstraps values greater than 50% were obtained in the analysis of 1000 replicates and are presented at the branching points. The tree was rooted with the A/goose/Guangdong/1/1996H5N1) HA sequence (NC_007362.1) as the outgroup.

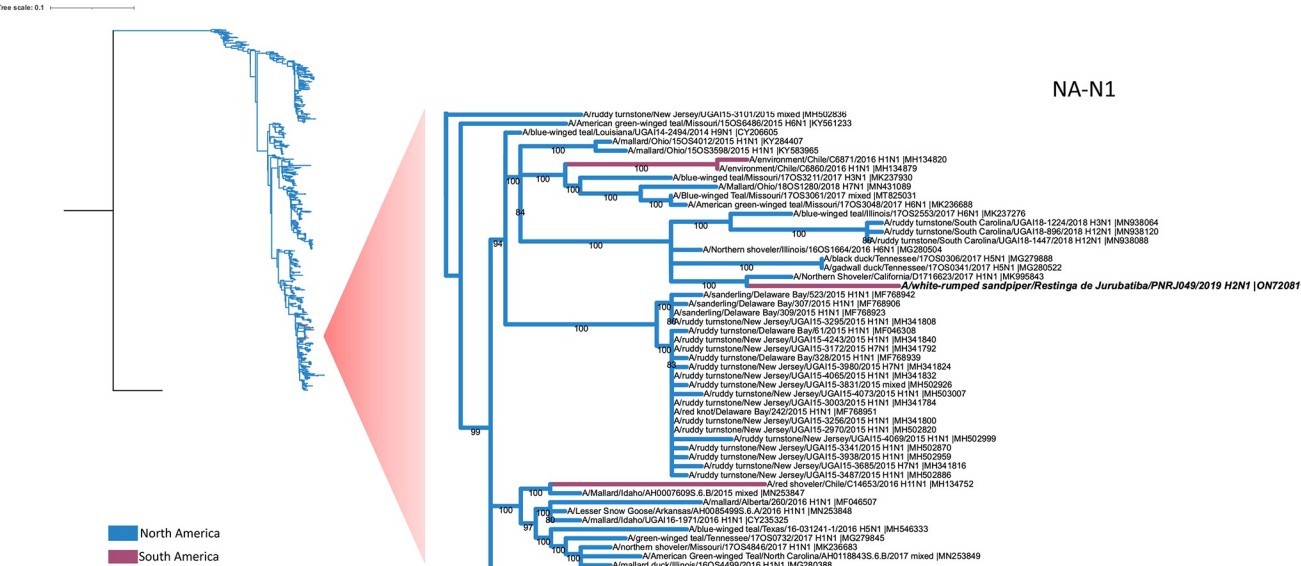

**Fig 3. Phylogenetic analysis of the PNRJ influenza isolate neuraminidase (NA-N1) gene.** The PNRJ sample is written in a bold font. North America branches are colored blue and South America are colored purple. The GTR+F+I substitution model was used as selected by IQ-TREE Model Finder, in a 1410 nt length alignment. The scale bar represents the number of substitutions per site. Bootstraps values greater than 50% were obtained in the analysis of 1000 replicates and are presented at the branching points. The tree was rooted with the A/District_Of_Columbia/50/2022(H1N1) NA sequence (OQ203115) as the outgroup.

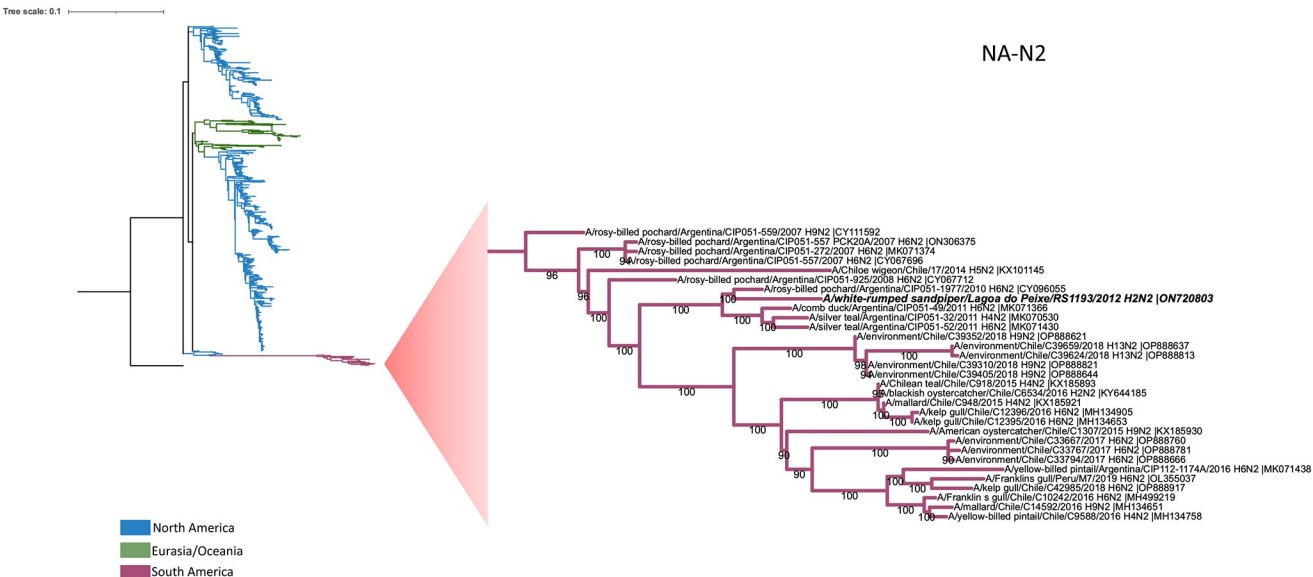

**Fig 4. Phylogenetic analysis of PNLP influenza isolate neuraminidase (NA-N2) gene.** The PNRJ sample is written in a bold font. North America branches are colored blue, Eurasia and Oceania are colored green and South America are colored purple. The scale bar represents the number of substitutions per site. The GTR+F+I substitution model was used as selected by IQ-TREE Model Finder, in a 1410 nt length alignment. Bootstraps values greater than 50% were obtained in the analysis of 1000 replicates and are presented at the branching points. The tree was rooted with the A/Korea/426/1968 (H2N2) NA2 sequence (NC_007382) as the outgroup.

Based on the identity at the nucleotide level analysis, the A(H2N2) virus showed its HA most closely with that A/lesser black-backed gull/Iceland/1597/2012 (H2N7) with 98.51% identity (CY149380.1).

Its NA grouped, with high bootstrap support and up to 99.29% sequence identity, with other viruses isolated in South America (Argentina, Chile and Peru) suggesting a regionally circulating lineage (CY096055.1; KY644185.1; OL355037.1) (Fig 4); similar trends were seen with the M (MK071423.1) and NP (KX620073.1) segments (supplementary material). Other gene segments of the A(H2N2) virus were most similar to viruses detected in North America. The NS (KX620095.1) and NP (KX620073.1) segments were most similar to Brazilian viruses also collected from Lagoa do Peixe National Park in 2012. Analysis of the deduced HA protein sequence showed a low pathogenic cleavage site identical to the A(H2N1) virus (VPQIESRGLF).

## Discussion

In this study, we describe two H2 subtype AIV collected from wild birds found in different sites in Brazil. Both viruses were detected in the same species (*C. fuscicolis*) but were genetically distinct (Fig 2). The white-rumped sandpiper (*C. fuscicolis*) is a coastal small shorebird which breeds in the northern tundra of Canada and Alaska. They are a long distance migrant, wintering in southern South America and the Caribbean. Our study suggests that *C. fuscicolis* frequenting the Brazilian Atlantic coast are a reservoir of AIVs in South America.

Previous studies of AIVs in Brazil, identified an A(H2N1) virus in a semi-palmated sandpiper (*Calidris pusilla*), Coroa do Avião island, in the northeast of Brazil [Lat. -7.817851 Long. -34.834895] in 1990 (Obenauer 2006) and an A(H2N2) virus reported in *C. fuscicollis* in 2012 (Araujo 2018). Prior to this study, complete sequence data were available for only three South

American A(H2) viruses, the A(H2N1) from Brazil in 1990, an A(H2N9) virus isolated from a duck in Peru in 2008 [27] and an A(H2N2) virus from a blackish oystercatcher (*Haematopus ater*) in Chile in 2016 [28].

Monitoring the biological and genetic characteristics of influenza viruses, such as the H2 viruses described here, present in the migratory bird populations is important as they have the potential to spread over long distances and to be exposed to numerous hosts. It is worth mentioning that viruses of the H2 subtype have already caused a pandemic in humans. This virus was generated by reassortment of the previously circulating human A(H1N1) virus and an avian A(H2N2) virus, culminating in the loss of million of lives [29].

Interestingly, although collected from the same host, the two viruses that we sequenced had gene segments, particularly PB2 and PB1, that were phylogenetically distant showing considerable viral genetic diversity might be present within this host. Further targeted surveillance of this host throughout its migratory route would shed light on this possibility and whether the viruses detected are maintained long term or simply introduced from other hosts. The fact that some gene segments, including the NP from the A(H2N2) virus belonged to a specific South American lineage does suggest that these viruses, or at least gene segments, may have been picked up locally. The presence of gene segments more related to viruses from North America does, however, complicate this hypothesis and clearly viruses move with some freedom between the continents. This is supported by the high degree of genetic similarity between genes of the A(H2N2) virus to those of an AIV identified in Iceland in the same year [30]. Thus, it is likely that new viral diversity is introduced into the Brazilian bird populations each year, affording the opportunity for reassortment with local viruses. Whether such viruses can move back from South to North America is not clear. There are still many gaps to be filled in our understanding of viral gene flow between viruses circulating in the two hemispheres, but our work contributes to understanding of the importance of the Atlantic route as a corridor for the movement of AIVs between North America, South America and even Europe.

## Supporting information

**S1 Fig. Phylogenetic analysis of influenza isolates polymerase basic 2 (PB2) segment.** Brazilian samples in this study are written in a bold font. North America branches are colored blue. The GTR+F+I substitution model was used as selected by IQ-TREE Model Finder, in a 2280 nt length alignment. The scale bar represents the number of substitutions per site. Bootstraps values greater than 50% were obtained in the analysis of 1000 replicates and are presented at the branching points. The tree was rooted with the A/Korea/426/1968(H2N2) PB2 sequence (NC_007378) as the outgroup.
(PDF)

**S2 Fig. Phylogenetic analysis of influenza isolates polymerase basic 1 (PB1) segment.** Brazilian samples in this study are written in a bold font. North America branches are colored blue, Iceland are colored gold and South America are colored purple. The GTR+F+I substitution model was used as selected by IQ-TREE Model Finder, in a 2317 nt length alignment. The scale bar represents the number of substitutions per site. Bootstraps values greater than 50% were obtained in the analysis of 1000 replicates and are presented at the branching points. The tree was rooted with the A/Korea/426/1968(H2N2) PB1 sequence (NC_007375) as the outgroup.
(PDF)

**S3 Fig. Phylogenetic analysis of influenza isolates nucleoprotein (NP) segment.** Brazilian samples in this study are written in a bold font. North America branches are colored blue and

South America are colored purple. The GTR+F+I substitution model was used as selected by IQ-TREE Model Finder, in a 1497 nt length alignment. The scale bar represents the number of substitutions per site. Bootstraps values greater than 50% were obtained in the analysis of 1000 replicates and are presented at the branching points. The tree was rooted with the A/Korea/426/1968(H2N2) NP sequence (NC_007381) as the outgroup.
(PDF)

**S4 Fig. Phylogenetic analysis of influenza isolates matrix (M) segment.** Brazilian samples in this study are written in a bold font. North America branches are colored blue and South America are colored purple. The SYM+I+I substitution model was used as selected by IQ-TREE Model Finder, in a 989 nt length alignment. The scale bar represents the number of substitutions per site. Bootstraps values greater than 50% were obtained in the analysis of 1000 replicates and are presented at the branching points. The tree was rooted with the A/Korea/426/1968(H2N2) Matrix sequence (NC_007377) as the outgroup.
(PDF)

**S5 Fig. Phylogenetic analysis of influenza isolates polymerase acidic (PA) segment.** Brazilian samples in this study are written in a bold font. North America branches are colored blue, Iceland are colored gold and South America are colored purple. The GTR+F+I substitution model was used as selected by IQ-TREE Model Finder, in a 2151 nt length alignment. The scale bar represents the number of substitutions per site. Bootstraps values greater than 50% were obtained in the analysis of 1000 replicates and are presented at the branching points. The tree was rooted with the A/Korea/426/1968(H2N2) PA sequence (NC_007376) as the outgroup.
(PDF)

**S6 Fig. Phylogenetic analysis of influenza isolates nonstructural (NS) segment.** Brazilian samples in this study are written in a bold font. North America branches are colored blue and South America are colored purple. The TVM+F+G substitution model was used as selected by IQ-TREE Model Finder, in an 837 nt length alignment. The scale bar represents the number of substitutions per site. Bootstraps values greater than 50% were obtained in the analysis of 1000 replicates and are presented at the branching points. The tree was rooted with the A/Korea/426/1968(H2N2) NS sequence (NC_007380) as the outgroup.
(PDF)

## Author Contributions

**Conceptualization:** Luciano M. Thomazelli, Erick G. Dorlass, Tatiana Ometto, Carla Meneguin, Robert G. Webster, Richard J. Webby, Edison L. Durigon, Jansen de Araujo.

**Data curation:** Luciano M. Thomazelli, Thomas Fabrizio, Maria Virgínia Petry.

**Formal analysis:** Erick G. Dorlass, Danielle Paludo, Robert G. Webster.

**Funding acquisition:** Luciano M. Thomazelli, João Renato Rebello Pinho, Richard J. Webby, Edison L. Durigon, Jansen de Araujo.

**Investigation:** Luciano M. Thomazelli, Tatiana Ometto, Carla Meneguin, David Walker, Scott Krauss, Thomas Fabrizio, Angelo L. Scherer, Jansen de Araujo.

**Methodology:** Luciano M. Thomazelli, João Renato Rebello Pinho, Erick G. Dorlass, Tatiana Ometto, Carla Meneguin, Danielle Paludo, Rodolfo Teixeira Frias, Patricia Luciano Mancini, Cairo Monteiro, Sophie Marie Aicher, David Walker, Guilherme P. Scagion, Thomas

Fabrizio, Maria Virgínia Petry, Angelo L. Scherer, Janete Scherer, Patricia P. Serafini, Isaac S. Neto, Deyvid Emanuel Amgarten, Fernanda de Mello Malta, Ana Laura Boechat Borges, Jansen de Araujo.

**Project administration:** Jansen de Araujo.

**Software:** Erick G. Dorlass, Guilherme P. Scagion, Deyvid Emanuel Amgarten.

**Supervision:** Luciano M. Thomazelli, Scott Krauss, Thomas Fabrizio, Richard J. Webby, Edison L. Durigon.

**Validation:** Erick G. Dorlass, David Walker.

**Writing – original draft:** Luciano M. Thomazelli, Erick G. Dorlass, Tatiana Ometto, Thomas Fabrizio, Richard J. Webby, Edison L. Durigon, Jansen de Araujo.

**Writing – review & editing:** Luciano M. Thomazelli, João Renato Rebello Pinho, Erick G. Dorlass, Danielle Paludo, Robert G. Webster, Richard J. Webby, Edison L. Durigon, Jansen de Araujo.

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
