## [Decision Letter · Decision Letter 0]

20 Sep 2023

PONE-D-23-19373Evidence of reassortment of A (H2) avian influenza viruses in shorebirds in BrazilPLOS ONE

Dear Dr. de Araujo,

Thank you for submitting your manuscript to PLOS ONE. After careful consideration, we feel that it has merit but does not fully meet PLOS ONE’s publication criteria as it currently stands. Therefore, we invite you to submit a revised version of the manuscript that addresses the points raised during the review process.

We look forward to receiving your revised manuscript.

Kind regards,

James Lee Crainey, Ph.D.

Academic Editor

PLOS ONE

“This work was supported by the Fundação de Amparo à Pesquisa do Estado de São Paulo, Grant/Award Number: 2011/13821-7 and  2017/01125-2; US National Institute of Allergy and Infectious Disease Centers of Excellence for Influenza Research and Surveillance (CEIRS) and by the American Lebanese Syrian Associated Charities (ALSAC) program, Grant/Award Number: (HHSN266200700005C); Wildlife Conservation Society (WCS) Project:, Grant/ Award Number: 2008005_WCS_OWOH and 2009005_WCS_OWOH; Fundação de Amparo à Pesquisa do Estado do Rio Grande do Sul, Grant/Award Number: 09/0574-7. Field campaigns on RJNP were supported by ICMBio through the GEF Mar Project for monitoring shorebirds.”

6. Please upload a new copy of Figure 3 as the detail is not clear. Please follow the link for more information: https://blogs.plos.org/plos/2019/06/looking-good-tips-for-creating-your-plos-figures-graphics/" https://blogs.plos.org/plos/2019/06/looking-good-tips-for-creating-your-plos-figures-graphics/

Reviewers' comments:

Reviewer's Responses to Questions

**Comments to the Author**

1. Is the manuscript technically sound, and do the data support the conclusions?

Reviewer #1: Partly

Reviewer #2: Yes

2. Has the statistical analysis been performed appropriately and rigorously? 

Reviewer #1: N/A

Reviewer #2: I Don't Know

3. Have the authors made all data underlying the findings in their manuscript fully available?

Reviewer #1: Yes

Reviewer #2: No

4. Is the manuscript presented in an intelligible fashion and written in standard English?

Reviewer #1: Yes

Reviewer #2: Yes

5. Review Comments to the Author

Reviewer #1: Thomazalli: Evidence of reassortment of A (H2) avian influenza viruses in shorebirds in Brazil.

The authors report the phylogenetic results of two H2 viruses found in waders in South America.

I find the framing of this study strange. H2 is a common LPAI subtype in wild birds, and so is framing LPAI viruses from shorebirds as a human pandemic risk really logical? Shouldn’t this be framed in the context of avian influenza and LPAI in general?

Along the same vein, the discussion is very locally focussed and doesn’t draw any information from global dynamics. For example, that H2 viruses from Terns in Australia were similarly mosaic viruses seems interesting context.

Line 67. Do you really mean that these disease signs are found in waterfowl (And without a reference)? In my experience, H2 is routinely detected and isolated from health waterfowl. I think you mean poultry?

Line 68. “variants” is not the correct terminology at all. You mean “subtypes”.

Line 78. This general phylogenetic division is found in most AIV subtypes, and this should be mentioned. I would say something like: Like most AIV subtypes, H2 is phylogenetic divided into two broad clades, based on geographic circulation. Unlike most AIV subtypes, however, North American sequences can be found in both the classic North America and Eurasian clades as a result of a transhemispheric transmission event and subsequent proliferation.” Line 80-81 is too vague.. what do you mean “by documented”. More detail.

Missing entirely is the role of shorebirds. Missing entirely is the interesting history of geographic mosasism, very much limited to gulls, terns and shorebirds. This is more relevant for this paper, than H2 human pandemic risk.

Line 104. Do you mean combined cloacal oral swabs, OR either cloacal or oral swabs were collected?

Line 114. Which VTM formulations?

Line 152-157. More detail about which tools and algorithms are involved here would be useful.

Lin e164. Why did you use the GTR+G+I for all segments. This is not in line with the best fit models for all segments, and nucleotide sub models have an impact on phylogenetic structure. Just choosing the most complex isn’t necessarily the best one.

Line 167-169 is repitition of line 164. Lines 169-175 are not required.

Line 177. How did you remove gaps? Manually? Did you maybe delete a base due to poor quality gappy sequences?

Line 185-188. More detail. How many samples from which species? Did you target the same populations repeatedly? How does prevalence change with time or species?

Figure 1. The pacific flyway does not include Florida. Birds also don’t really fly over the open from Florida all to the way to Argentina. The Atlantic flyway doesn’t start in the open ocean. I think that the flyway part of this figure needs to be reconsidered. I appreciate that its conceptual.. but it could be improved substnaitaly. The text on top of the photos is not legible. Currently not an effective figure.

Line 200. Subjected is not the correct word… Rather you generated sequences from these samples.

Line 212: Pls add the GenBank Accession for the virus from Wisconsin, from Ohio, from California etc

Table 1. The influenza designations are incorrect. Why is the sample ID at the end? It should be A/host/Country/SampleID/Year(HxNx). Please fix this. Verify that it is correct in all figures and the text.

Figure 2. The small sub-tree is too small. I cant read the tip labels so this needs to be resolved. The bright green is impossible to read.

Can you add branch colours to the big tree too?

There is major rooting issue on the mega tree that needs to be resolved.

You need to specify in the legend how you rooted the tree.

What sequences are in this big tree? All H2 viruses in GenBank? Or for only certain time frames?

And “branch lengths proportional to evolutionary distance” is vague. Should this not be “scale bar represes number of substitutions per site”? Or is IQ tree doing something different?

Figure 3/4.

Please clarify which sequences are in the mega tree. Colour branches. Rooting issues here (Fig 3, Fig 4 looks ok). Please clarify rooting. Please add a scale bar.

Change NA1/NA2 to N1/N2 OR NA-N1/NA-N2.

The quality is low and tip labels grainy (Figure 3, but 4 is ok). Tips colorus are inconsistent.. shoudlnt all tips be coloured. The neon green is impossible to read. Where is the scale bar?

Why is the way the format of Figure 2 different from 3/4?

Supplemental figures consistently missing scale bars. The rooting on the large trees should be carefully assessed. There are inconsistencies in branch colouring. For some (e.g. the M tree) the tip names are not legible due to being tiny… so formatting needs to be fixed here.

Reviewer #2: I have reviewed the manuscript “Evidence of reassortment of A{H2} avian influenza viruses in shorebirds in Brazil”. The authors isolated and sequenced two H2 isolates and characterized them phylogenetically, documenting that the viral genomes were derived from a variety of geographic sources via reassortment. The paper used valid viral, genomic and phylogenetic techniques but the manuscript needs some revision, primarily to edit grammar and writing issues. I have listed several suggestions below but strongly suggest editorial review by the authors, perhaps enlisting experienced assistance.

I suggest editing the title to “Evidence of reassortment of avian influenza A {H2} viruses in Brazilian shorebirds”. This is more in line with accepted virus nomenclature.

Line 63, 64 probably needs a citation citing the “concern”. Maybe Joseph et al. 2015.

Line 75. Remove “countries including”

Line 83. I am not sure how this surveillance was “prospective” particularly since samples were collected in part, in 2012.

Line 84. Sentence needs revision as it contains 4 “ins” within 9 words.

Line 92. Can delete “however”

Line 100. Should maintain past tense so change to “provided”.

Lines 104 and 112. Authors use cloacal/oral and then oropharyngeal/cloacal to describe sampling technique. Which is it and be consistent. Also, line 104 swabs should be singular.

Line 114. Does the composition of VTM need to be defined?

Line 133. Virus isolation “was” attempted…

Delete “by standard methods” as the authors then describe technique and provide citation.

Line 152. Bioinformatics should be singular. Remove “result”

Line 153. Capitalize “Varsmetagen”

Lines 163 and 167. Do these reference the same thing? If so then this is repetitive.

Line 170. Should be Neighbor Joining not join

Line 185. Out of 1212 swab samples only two were positive? This seems low conceptually. Were other viruses detected and if so, how many and were the H2 viruses the only ones characterized?

Line 185 should read “a total OF 1212”

Line 225. Need scientific name after species.

Line 253. Should read …are a reservoir…

Line 265. Need scientific name

Line 273. I found estimates of 1.3 million lives lost, not millions.

Line 276. “a lot of” seems jargony. Maybe use “considerable” or some other word instead.

Line 288. Should use a citation for the Iceland work. Dusek et al. 2014.

Line 293. Needs a word after gene. Viral gene what?

Table 1 can be eliminated as it was presented in the text.

Figure legends can be revised. I suggest revising the first sentences of all figure legends to read more like “Phylogenetic analysis of the PNRJ influenza isolate HA gene” or similar. This would read better.

Figure 1 legend should be “are indicated”

6. PLOS authors have the option to publish the peer review history of their article (what does this mean?). If published, this will include your full peer review and any attached files.

Reviewer #1: No

Reviewer #2: No

---

## [Author Response · Author response to Decision Letter 0]

30 Nov 2023

To the Academic Editor 

Dear James Lee Crainey, Ph.D, 

Editor’s report:

PONE-D-23-19373: Evidence of reassortment of A (H2) avian influenza viruses in shorebirds in Brazil

PLOS ONE

Thank you for considering our work for potential publication in PLOS ONE. I appreciate the comments of the reviewers about our study in Brazil. We agree with all of the suggestions and believe they result in a better understanding of the work. We have complied with the suggestions, as shown in the attached, new manuscript version. 

We look forward to your response, 

Sincerely yours,

Jansen de Araujo, Ph.D

Institute of Biomedical Science

University of São Paulo State, Brazil

Prof. Lineu Prestes Avenue, 1374, São Paulo.

E-mail: jansentequila@usp.br

Reviewer #1: 

 The authors report the phylogenetic results of two H2 viruses found in waders in South America. I find the framing of this study strange. H2 is a common LPAI subtype in wild birds, and so is framing LPAI viruses from shorebirds as a human pandemic risk really logical? Shouldn’t this be framed in the context of avian influenza and LPAI in general?

Answer: We totally agree that H2N2 is a LPAI found in wild birds. As we know the avian H2 virus is different genetically from the pandemic H2. However, the human H2N2 subtype disappeared after 1968, leaving no trace in the human population. Furthermore, unlike other places in the world, data on the existence of H2N2 subtypes circulating in wild birds in South America are practically non-existent. It is known that the LPAI virus is a precursor of the HPAI virus. In Brazil, the agribusiness system has increasingly advanced in green areas, and human contact has become increasingly close and frequent. The text was reformulated to improve understanding.

We included new sentence in discussion “…There are still many gaps to be filled in our understanding of viral gene flow between the two hemispheres, but demonstrates the importance of the Atlantic route as a corridor for the movement of AIVs between North America, South America and even Europe. This work came across with the scarcity of information on AIVs in South America emphasizing the necessity for additional studies on AIVs at the regional level, with a specific focus on identifying the endemic subtypes present in wild birds within the region…”

Along the same vein, the discussion is very locally focussed and doesn’t draw any information from global dynamics. For example, that H2 viruses from Terns in Australia were similarly mosaic viruses seems interesting context.

Answer: Thank you for the excellent suggestion. We reviewed our data and included new sequences from Australia and Eurasian in the phylogenetic tree analyses (Fig 1). 

Line 67. Do you really mean that these disease signs are found in waterfowl (And without a reference)? In my experience, H2 is routinely detected and isolated from health waterfowl. I think you mean poultry?

Answer: We have included a new reference to this sentence “Avian A(H2) viruses in wild birds typically exhibit no pathogenicity, but they can cause symptoms like sneezing, coughing, and nasal discharge in poultry and sinusitis in waterfowl” Reference: Reneer ZB, Ross TM. H2 influenza viruses: designing vaccines against future H2 pandemics. Biochem Soc Trans 2019;47: 251–264.

Line 68. “variants” is not the correct terminology at all. You mean “subtypes”.

Answer: We changed on text “Numerous subtypes of avian A(H2) viruses, including H2N1, H2N3, H2N5, H2N7, H2N8, and H2N9, have been reported in various animal hosts”

Line 78. This general phylogenetic division is found in most AIV subtypes, and this should be mentioned. I would say something like: Like most AIV subtypes, H2 is phylogenetic divided into two broad clades, based on geographic circulation. Unlike most AIV subtypes, however, North American sequences can be found in both the classic North America and Eurasian clades as a result of a transhemispheric transmission event and subsequent proliferation.”

Answer: Thanks for this point. As a suggestion, we added this sentence in our text “Like most AIV subtypes, H2 is phylogenetically divided into two broad clades, based on geographic circulation. Unlike most AIV subtypes, however, North American sequences can be found in both classic North America and Eurasian clades as a result of a trans-hemispheric transmission event and subsequent proliferation”

Line 80-81 is too vague.. what do you mean “by documented”. More detail.

Answer: We removed this sentence.

Missing entirely is the role of shorebirds. Missing entirely is the interesting history of geographic mosasism, very much limited to gulls, terns and shorebirds. This is more relevant for this paper, than H2 human pandemic risk.

Answer: Thanks for the suggestion. We added a new sentence about these wild birds to the text “Some Nearctic migratory species, breeds in the Canadian Arctic from May to June and migrates to Argentine and Chilean Patagonia between January and April [14]. During its southbound migration, birds enter Brazil via the northeast coast or the mouth of the Amazon River, possibly using areas in the Pantanal Biome[15]. They arrive on the southern coast of Brazil between November and January where keep together with several species such as gulls, terns, and shorebirds with direct contact, refuel, and then continue south to Patagonia. Their extensive migratory routes and susceptibility to viruses emphasize their role as transcontinental vectors of avian viruses.”

Line 104. Do you mean combined cloacal oral swabs, OR either cloacal or oral swabs were collected?

Answer: We corrected the sentence “…Combined cloacal/tracheal swabs samples were collected from wild birds (mainly from Charadriiformes) including long and intermediate migratory waterfowl and local resident…”

Line 114. Which VTM formulations?

Answer: We added new sentence to VTM formulations “The swabs were placed in vials containing VTM transport medium (VTM: PBS + 10% glycerol + Antibiotics/fungizone) and immediately placed in liquid nitrogen following collection”.

Line 152-157. More detail about which tools and algorithms are involved here would be useful.

Answer: Thank you for the suggestion. More details were added to the text: “The Varsmetagen virome pipeline (https://varsomics.com/varsmetagen/) performed the following steps: raw sequence metrics and filtering of low quality reads (sequences with length lower than 50 bp and Phrep score lower than 20); host decontamination by mapping reads to a close host genome (Gallus gallus assembly: GCF_016699485.2) using BWA with default parameters; first round of pathogen identification using Kraken2 [20] with a custom database including Virus, Bacteria and Human on NCBI database and complete viral genomes available in Genbank with a custom 33-kmer length; short reads assembly with Spades (version 3.13) [21]; second round of identification with contigs; search for distant homologous sequences using Hidden Markov Models (HMM) of viral proteins implemented in the eegNOG database [22] and finding confirmation through mapping and coverage metrics.”

Lin e164. Why did you use the GTR+G+I for all segments. This is not in line with the best fit models for all segments, and nucleotide sub models have an impact on phylogenetic structure. Just choosing the most complex isn’t necessarily the best one.

Answer: Thanks for suggestion. We re-run IQ-Tree with the “ModelFinder” parameter that search and uses the best model for each dataset. This information was corrected in the text. “Maximum Likelihood analysis was conducted using IQ-TREE [24] for all 8 AIV segments of each virus. The substitution model was chosen with the Model Finder parameter for each dataset.” Each substitution model selected by IQ-TREE Model Finder has been presented in a new version in each figure description.

Line 167-169 is repitition of line 164. 

Answer: The sentence was reformulated and combined in new version. “Maximum Likelihood analysis was conducted using IQ-TREE [24] for all 8 AIV segments of each virus. The substitution model was chosen with the Model Finder parameter for each dataset. Bootstrap was set to 1000 for statistical significance. Tree editing was performed with iTOL [25,26]. The tree was drawn to scale, with branch lengths measured in the number of substitutions per site. Codon positions included were 1st+2nd+3rd+Noncoding. Alignment was checked with Geneious Prime software, and all positions containing gaps and missing data were eliminated.”

Lines 169-175 are not required.

Answer: The sentence was removed as suggested “Initial tree(s) for the heuristic search were obtained automatically by applying Neighbor-Join and BioNJ algorithms to a matrix of pairwise distances estimated using the Maximum Composite Likelihood (MCL) approach and then selecting the topology with a superior log-likelihood value. A discrete Gamma distribution was used to model evolutionary rate differences among sites. The tree was drawn to scale, with branch lengths measured in the number of substitutions per site”.

Line 177. How did you remove gaps? Manually? Did you maybe delete a base due to poor quality gappy sequences?

Answer: The sequence alignment was double-checked with Geneious program. This sentenced was added: “Alignment was checked with Geneious Prime software, and all positions containing gaps and missing data were eliminated.” No base was deleted.

Line 185-188. More detail. How many samples from which species? Did you target the same populations repeatedly? How does prevalence change with time or species?

Answer: A total of 1212 birds were sampled at Lagoa do Peixe National Park (PNLP), belonging to different families including Ardeidae, Charadriidae, Haematopodidae, Recurvirostridae, Laridae, Rostratulidae, Tyrannidae, Furnariidae and Scolopacidae. The predominant family was Scolopacidae, with the majority being C. fuscicolis, accounting for 370 samples (30%). We aimed to capture distinct populations, and no animals were resampled throughout the entire collection period. In this study, we identified only one C. fuscicollis carrying the H2N2 subtype, representing a prevalence of 0.3% (1/370) in extreme South of Brazil. At the second site, Restinga de Jurubatiba National Park (RJNP), sampling was conducted in 2019, resulting in a total of 118 samples representing families Ardeidae, Charadriidae, Tyrannidae, Rostratulidae, Caprimulgidae, Anatidae, Motacillidae, Jacanidaea and Scolopacidae with the majority also belonging to the same species, forty-eight C. fuscicollis (representing 40%). All samples tested negative for avian influenza viruses, except for a single case of H2N1. Data on AIV prevalence in Brazil are still limited. So, only two wild birds (from the same species) were detected carrying the H2 subtype in South America. This unprecedented characterization is what we are presenting in the current study.

A new sentence was reformulated on text to clarify “A total 1212 oropharyngeal/cloacal samples were collected from wild birds at Lagoa do Peixe National Park (PNLP) during 2012, belonging to different families including Ardeidae, Charadriidae, Haematopodidae, Recurvirostridae, Laridae, Rostratulidae, Tyrannidae, Furnariidae and Scolopacidae. The predominant family was Scolopacidae, with the majority being C. fuscicolis, accounting for 370 samples (30%). We aimed to capture distinct populations, and no animals were resampled throughout the entire collection period. In this study, we identified only one C. fuscicollis carrying the H2N2 subtype, representing a prevalence of 0.3% (1/370) in extreme South of Brazil. At the second site, Restinga de Jurubatiba National Park (RJNP), sampling was conducted in 2019, resulting in a total of 118 samples representing families Ardeidae, Charadriidae, Tyrannidae, Rostratulidae, Caprimulgidae, Anatidae, Motacillidae, Jacanidaea and Scolopacidae with the majority also belonging to the same species, forty-eight C. fuscicollis (representing 40%). All samples tested negative for avian influenza viruses, except for a single case of H2N1 (Figure 1). Only two samples, one from PNLP and one from RJNP, were positive for H2 subtype, both from free-ranging white-rumped sandpipers (C. fuscicollis).”

Figure 1. The pacific flyway does not include Florida. Birds also don’t really fly over the open from Florida all to the way to Argentina. The Atlantic flyway doesn’t start in the open ocean. I think that the flyway part of this figure needs to be reconsidered. I appreciate that its conceptual.. but it could be improved substnaitaly. The text on top of the photos is not legible. Currently not an effective figure.

Answer: Thanks for pointing out. The figure was improved to better understand.

Line 200. Subjected is not the correct word… Rather you generated sequences from these samples.

Answer: The sentence was reformulated “Both samples underwent NGS sequencing, the sequence for RJNP049 was generated from the isolated virus and PNLP1193 was generated directly from the original sample.”

Line 212: Pls add the GenBank Accession for the virus from Wisconsin, from Ohio, from California etc

Answer: The GenBank accession numbers were included.

Table 1. The influenza designations are incorrect. Why is the sample ID at the end? It should be A/host/Country/SampleID/Year(HxNx). Please fix this. Verify that it is correct in all figures and the text.

Answer: Sorry about this, the table was removed as suggested by review 2.

Figure 2. The small sub-tree is too small. I cant read the tip labels so this needs to be resolved. The bright green is impossible to read.

Answer: Thank you for the suggestion. Indeed, the bright green was a poor choice for coloring the branches. The colors were changed for better viewing.

Can you add branch colours to the big tree too?

Answer: Yes, the Trees were reconstructed with all the suggestions

There is major rooting issue on the mega tree that needs to be resolved.

Answer: Yes, the Trees were reconstructed with all the suggestions

You need to specify in the legend how you rooted the tree.

Answer: Yes, the rooting specifications are now mentioned in the figure description.

What sequences are in this big tree? All H2 viruses in GenBank? Or for only certain time frames?

Answer: We used all H2 with complete cds available in NCBI Influenza Virus Database, totalizing 780 complete sequences.

And “branch lengths proportional to evolutionary distance” is vague. Should this not be “scale bar represes number of substitutions per site”? Or is IQ tree doing something different?

Answer: Ok, thanks for the suggestion. All legends were reformulated and included “The scale bar represents the number of substitutions per site”.

Figure 3/4. Please clarify which sequences are in the mega tree. Colour branches. Rooting issues here (Fig 3, Fig 4 looks ok). Please clarify rooting. Please add a scale bar.

Answer: Thank you for the suggestions. The branches were colored for better clarification. The rooting was solved, and the scale bar was added in the new version.

Change NA1/NA2 to N1/N2 OR NA-N1/NA-N2.

Answer: Sorry to be disappointed. We corrected all in figures.

The quality is low and tip labels grainy (Figure 3, but 4 is ok). Tips colorus are inconsistent.. shoudlnt all tips be coloured. The neon green is impossible to read. Where is the scale bar?

Answer: Thank you for the suggestion, we are submitting the trees with correct branch coloring and scale bar. In the new version.

Why is the way the format of Figure 2 different from 3/4?

Answer: Thank you for pointing that out. We are resubmitting the trees with the format standardized as Figure 2.

Supplemental figures consistently missing scale bars. The rooting on the large trees should be carefully assessed. There are inconsistencies in branch colouring. For some (e.g. the M tree) the tip names are not legible due to being tiny… so formatting needs to be fixed here.

Answer: Thank you for the suggestion. We are resubmitting the trees with better coloring, tip sizes, and rooting descriptions.

Reviewer #2: 

I have reviewed the manuscript “Evidence of reassortment of A{H2} avian influenza viruses in shorebirds in Brazil”. The authors isolated and sequenced two H2 isolates and characterized them phylogenetically, documenting that the viral genomes were derived from a variety of geographic sources via reassortment. The paper used valid viral, genomic and phylogenetic techniques but the manuscript needs some revision, primarily to edit grammar and writing issues. I have listed several suggestions below but strongly suggest editorial review by the authors, perhaps enlisting experienced assistance.

Answer: Thank you for your constructive criticism to help us make this manuscript better. The lead author is currently working with the journal to ensure the figures are of adequate quality and readability for publication.

I suggest editing the title to “Evidence of reassortment of avian influenza A {H2} viruses in Brazilian shorebirds”. This is more in line with accepted virus nomenclature.

Answer: The suggestion was accepted. The title was changed “Evidence of reassortment of avian influenza A {H2} viruses in Brazilian shorebirds”

Line 63, 64 probably needs a citation citing the “concern”. Maybe Joseph et al. 2015.

Answer: Thanks for pointed. We added the reference as suggested.

Line 75. Remove “countries including”

Answer: The “countries including” was removed.

Line 83. I am not sure how this surveillance was “prospective” particularly since samples were collected in part, in 2012.

Answer: The “prospective” was removed.

Line 84. Sentence needs revision as it contains 4 “ins” within 9 words.

Answer: This sentence was reformulated, “Over the past 15 years, we have conducted influenza surveillance in wild birds at various sites in Brazil, including shorebirds at Lagoa do Peixe National Park (PNLP) and the Amazon region.”

Line 92. Can delete “however”

Answer: Ok. We agree, and removed “however” from the sentence.

Line 100. Should maintain past tense so change to “provided”.

Answer: The sentence was corrected.

Lines 104 and 112. Authors use cloacal/oral and then oropharyngeal/cloacal to describe sampling technique. Which is it and be consistent. Also, line 104 swabs should be singular.

Answer: The sentence was reformulated.

Line 114. Does the composition of VTM need to be defined?

Answer: We added new sentence to VTM formulations “The swabs were placed in vials containing VTM transport medium (VTM: PBS + 10% glycerol + Antibiotics/fungizone) and immediately placed in liquid nitrogen following collection”.

Line 133. Virus isolation “was” attempted…

Answer: The sentence was corrected. “…Virus isolations was attempted…”

Delete “by standard methods” as the authors then describe technique and provide citation.

Answer: The “by standard methods” was removed.

Line 152. Bioinformatics should be singular. Remove “result”

Answer: The sentence was corrected.

Line 153. Capitalize “Varsmetagen”.

Answer: The sentence was corrected “Varsmetagen”.

Lines 163 and 167. Do these reference the same thing? If so then this is repetitive.

Answer: The sentence was combined in new version. “Maximum Likelihood analysis was conducted using IQ-TREE [24] for all 8 AIV segments of each virus. The substitution model was chosen with the Model Finder parameter for each dataset. Bootstrap was set to 1000 for statistical significance. Tree editing was performed with iTOL [25,26]. The tree was drawn to scale, with branch lengths measured in the number of substitutions per site. Codon positions included were 1st+2nd+3rd+Noncoding. Alignment was checked with Geneious Prime software, and all positions containing gaps and missing data were eliminated.”

Line 170. Should be Neighbor Joining not join

Answer: The sentence was reformulated. “We performed a phylogenetic analysis of the obtained IAV sequences to investigate their genetic relationship to other influenza virus sequences available in GenBank. Sequence alignments were performed using MAFFT [23]. Maximum Likelihood analysis was conducted using IQ-TREE [24] for all 8 AIV segments of each virus. The substitution model was chosen with the Model Finder parameter for each dataset. Bootstrap was set to 1000 for statistical significance. Tree editing was performed with iTOL [25,26]. The tree was drawn to scale, with branch lengths measured in the number of substitutions per site. Codon positions included were 1st+2nd+3rd+Noncoding. Alignment was checked with Geneious Prime software, and all positions containing gaps and missing data were eliminated.”

Line 185. Out of 1212 swab samples only two were positive? This seems low conceptually. Were other viruses detected and if so, how many and were the H2 viruses the only ones characterized?

Answer: A total of 1212 birds were sampled at Lagoa do Peixe National Park (PNLP), belonging to different families including Ardeidae, Charadriidae, Haematopodidae, Recurvirostridae, Laridae, Rostratulidae, Tyrannidae, Furnariidae and Scolopacidae. The predominant family was Scolopacidae, with the majority being C. fuscicolis, accounting for 370 samples (30%). We aimed to capture distinct populations, and no animals were resampled throughout the entire collection period. In this study, we identified only one C. fuscicollis carrying the H2N2 subtype, representing a prevalence of 0.3% (1/370) in extreme South of Brazil. At the second site, Restinga de Jurubatiba National Park (RJNP), sampling was conducted in 2019, resulting in a total of 118 samples representing families Ardeidae, Charadriidae, Tyrannidae, Rostratulidae, Caprimulgidae, Anatidae, Motacillidae, Jacanidaea and Scolopacidae with the majority also belonging to the same species, forty-eight C. fuscicollis (representing 40%). All samples tested negative for avian influenza viruses, except for a single case of H2N1. Data on AIV prevalence in Brazil are still limited. So, only two wild birds (from the same species) were detected carrying the H2 subtype in South America. This unprecedented characterization is what we are presenting in the current study.

A new sentence was reformulated on text to clarify “A total 1212 oropharyngeal/cloacal samples were collected from wild birds at Lagoa do Peixe National Park (PNLP) during 2012, belonging to different families including Ardeidae, Charadriidae, Haematopodidae, Recurvirostridae, Laridae, Rostratulidae, Tyrannidae, Furnariidae and Scolopacidae. The predominant family was Scolopacidae, with the majority being C. fuscicolis, accounting for 370 samples (30%). We aimed to capture distinct populations, and no animals were resampled throughout the entire collection period. In this study, we identified only one C. fuscicollis carrying the H2N2 subtype, representing a prevalence of 0.3% (1/370) in extreme South of Brazil. At the second site, Restinga de Jurubatiba National Park (RJNP), sampling was conducted in 2019, resulting in a total of 118 samples representing families Ardeidae, Charadriidae, Tyrannidae, Rostratulidae, Caprimulgidae, Anatidae, Motacillidae, Jacanidaea and Scolopacidae with the majority also belonging to the same species, forty-eight C. fuscicollis (representing 40%). All samples tested negative for avian influenza viruses, except for a single case of H2N1 (Figure 1). Only two samples, one from PNLP and one from RJNP, were positive for H2 subtype, both from free-ranging white-rumped sandpipers (C. fuscicollis).”

Line 185 should read “a total OF 1212”

Answer: The sentence was corrected.

Line 225. Need scientific name after species.

Answer: The sentence was added reference and scientific name. “..from Northern Shovelers (Spatula clypeata) in California (99.04%) and Illinois (98.76%) (MK995843.1). Genbank accession numbers.”

Line 253. Should read …are a reservoir…

Answer: The sentence was corrected.

Line 265. Need scientific name

Answer: The scientific name was added. “…virus from a blackish oystercatcher (Haematopus ater) in Chile in 2016…”

Line 273. I found estimates of 1.3 million lives lost, not millions.

Answer: The sentence was corrected.

Line 276. “a lot of” seems jargony. Maybe use “considerable” or some other word instead.

Answer: Thanks for suggestion. We changed in new version.

Line 288. Should use a citation for the Iceland work. Dusek et al. 2014.

Answer: The Reference was added in new version as suggested.

Line 293. Needs a word after gene. Viral gene what?

Answer: The sentence was corrected “…There are still many gaps to be filled in our understanding of viral gene flow between the two hemispheres…”

Table 1 can be eliminated as it was presented in the text.

Answer: The table was removed as suggested. 

Figure legends can be revised. I suggest revising the first sentences of all figure legends to read more like “Phylogenetic analysis of the PNRJ influenza isolate HA gene” or similar. This would read better.

Answer: Thanks for suggestions. All legends were reformulated.

Figure 1 legend should be “are indicated”

 Answer: Thanks for suggestions. The legend was corrected.

---

## [Decision Letter · Decision Letter 1]

14 Jan 2024

PONE-D-23-19373R1Evidence of reassortment of avian influenza A {H2} viruses in Brazilian shorebirdsPLOS ONE

Dear Dr. Jansen de Araujo

Thank you for submitting your manuscript to PLOS ONE. After careful consideration, we feel that it has merit but does not fully meet PLOS ONE’s publication criteria as it currently stands. Therefore, we invite you to submit a revised version of the manuscript that addresses the points raised during the review process.

Although I am again requesting minor corrections, I want to make clear that I believe the manuscript has progressed significantly and needs far fewer modifications than before and that I am not expecting it to need to be sent out to reviewers again. 

Regrettably, neither of the original reviewers accepted invitations to review your revised manuscript. After looking carefully at their comments and your revised manuscript I was unconvinced that all of the issues raised by referee 1 about the figure presentation and data analysis had been adequately delt with and thus chose to send it out to another reviewer. 

Referee 3 has, in my view, provided an excellent review of your revised paper as well as many good suggestions as to how you can deal with the figure presentation issues that they and referee 1 (previously) identified.

Reviewer 3 has also identified some errors in your interpretation of your phylogenetic analysis. These errors must be corrected for your article to comply with PLoS One publication guidelines so please provide detailed responses to all these comments that concern the phylogenetic analysis section of your results and make sure that all appropriate corrections are made.  

In my view, reviewer 3 has also provided very helpful advice which provides you with a clear path to the publication of your work in PLoS One. I, therefore, hope to see a revised version of your manuscript published in PLoS One in the very near future. 

We look forward to receiving your revised manuscript.

Kind regards,

James Lee Crainey, Ph.D.

Academic Editor

PLOS ONE

Journal Requirements:

Reviewers' comments:

Reviewer's Responses to Questions

**Comments to the Author**

1. If the authors have adequately addressed your comments raised in a previous round of review and you feel that this manuscript is now acceptable for publication, you may indicate that here to bypass the “Comments to the Author” section, enter your conflict of interest statement in the “Confidential to Editor” section, and submit your "Accept" recommendation.

Reviewer #3: (No Response)

2. Is the manuscript technically sound, and do the data support the conclusions?

Reviewer #3: Partly

3. Has the statistical analysis been performed appropriately and rigorously? 

Reviewer #3: No

4. Have the authors made all data underlying the findings in their manuscript fully available?

Reviewer #3: (No Response)

5. Is the manuscript presented in an intelligible fashion and written in standard English?

Reviewer #3: No

6. Review Comments to the Author

Reviewer #3: Thomazelli et al. report the occurrence and genomic sequences of two influenza A viruses from wild-caught white-rumped sandpipers (Calidris fuscicollis). One was an A(H2N1) virus isolated from a bird caught in Restinga de Jurubatiba National Park (PNRJ, Rio de Janeiro) and one was a A(H2N2) virus isolated from a wild caught bird inhabiting the Lagoa do Peixe National Park (PNLP, Rio Grande do Sul). DNA sequencing and phylogenetic analysis of the recovered sequences showed each to be from different subtypes. Phylogenetic analysis also showed that while all the genomic sequence recovered from PNRJ-isolated virus was most closely related to other A(H2N1) viruses isolated from North American birds, the A(H2N2) virus genome recovered from the PNLP captured bird contained some sequences that were most closely related to viruses recovered from Iceland and North America and others that were most closely related to virus sequences recovered from birds caught in South America.

I believe the manuscript is, in general, well-organized and written in good clear easy-to-follow English and I believe the data contained in the manuscript is valuable and will be considered a welcome contribution to the field. I believe the analysis used was appropriate for the study and that the key conclusions the authors have drawn are supported by the authors data and reasonable. However, many of figures are inadequately presented and some of the interpretation of the phylogenetic analysis is incorrect. Labelling of the virus sequences and referencing of viruses is also inconsistent which makes the article unnecessarily difficult to understand in places. Below I am providing detailed comments about how I think the authors can improve their manuscript and indeed resolve their phylogenetic interpretation issues so that the authors can produce a revised manuscript that I would be delighted to recommend for publication in PLoS One.

Abstract

Overall, I feel there is insufficient information about what exactly has been found. For example, there is no mention of the fact genomic sequences for these viruses have been determined and no direct mention of the results obtained from the author´s phylogenetic analysis (only a mention of the conclusions drawn from this analysis). As PLoS One allows 300-word abstracts and the current abstract is only 169 words so there is plenty of space to provide this detail. I recommend, thus, that they expanded their existing abstract to include additional details.

Line 37: First sentence. The word “potential is redundant and should be deleted.

Introduction:

Line 84: Although I understand what the authors intend to say here, I think clarifications are needed. I suggest the sentence “Unlike most AIV subtypes, however, North American sequences can be found in both classic North America and Eurasian clades as a result of a trans-hemispheric transmission event and subsequent proliferation” should be changed to: “Unlike most AIV subtypes, however, “North American” clade sequences can be found in both classic North America and Eurasian viral genomes as a result of a trans-hemispheric transmission event and subsequent proliferation”

Line 99: Breeds should be corrected to “breed” and “migrates” to “migrate”.

Line 104: The last two sentences of the penultimate paragraph of the introduction need to be reformulated for English language clarity:

“They arrive on the southern coast of Brazil between November and January where keep together with several species such as gulls, terns, and shorebirds with directly contact, refuel, and then continue south to Patagonia. Their extensive migratory routes and susceptibility to 108 viruses emphasize their role as transcontinental vectors of avian viruses.”

Lines 111-115. The first three sentences of the last paragraph of the introduction need to be revised for clarity. I suggest changing it to something like this:

In this study, two A(H2) viruses were isolated from wild-caught white-rumped sandpipers (Calidris fuscicollis). One was an A(H2N1) virus obtained from a bird caught in Restinga de Jurubatiba National Park (PNRJ, 113 Rio de Janeiro); the other, an A(H2N2) virus, was isolated from a bird caught in Lagoa do Peixe National Park (PNLP, Rio Grande do Sul) in the extreme south of Brazil.

Fig 1. There seems to be no reference to the bird migration routes shown in figure 1. I suggest adding a “(see figure 1)” to the end of the last sentence of the penultimate paragraph and providing some description of these routes in the figure caption. Alternatively, the authors could revise the figure (removing this information) and provide more information on the collection site, if this is the only purpose the figure is to serve.

Methods

Phylogenetic analysis section

Line 19: It appears that the authors used nucleotide sequence alignments (not inferred protein sequences) for the phylogenic analysis; however, they don´t make this explicit. They also state they used 8 AIV segments but do not provide the sequence alignments that they used for these trees. Including the sequence alignment would allow others to build on their study and so I recommend that they are all included in the supplementary material. Failing this, I think the authors should at least provide information of the length of the sequence alignment used to construct each of their phylogenetic trees.

Line 192: the sentence “Bootstrap was set to 1000 for statistical significance” needs to be revised for accuracy/clarity. I suggest changing to something like: “The statistical significance of phylogenetic groupings was tested with bootstrap analysis using 1000 replicates”.

Results

Sampling

Line 218: Presumably all samples were tested with the RT-PCR described in the methods section; however, I feel I would be helpful to make this explicit here in the results section.

Line 219: The reference to figure 1 at this juncture does not make sense, please delete.

Phylogenetic analysis section

Line 223: Presently reads thus: “Phylogenetic analysis of the A(H2N1) virus showed that its internal genes clustered not with those from viruses in South America, but with those of influenza viruses isolated from North America (Fig 2 and 3)”.

I think this should be revised for clarity to something like: “Phylogenetic analysis of the A(H2N1) virus recovered from the PNRJ capture bird showed that all of its internal genes clustered not with gene sequence isolated from birds sampled in South America, but with sequences of influenza viruses isolated from birds sampled in North America (Fig 2, 3, SF1-5)

Lines 241 to 250: The information provided here is not phylogenetic analysis or helpful for interpreting the authors data. I strongly recommend that it is deleted as some of it is, in fact, miss-leading. For example, a correct interpretation of the phylogenetic analysis shown in supplementary figure 5 is that the PA gene of A(H2N1) virus from PNRJ is equally closely related to a Mississippi isolated virus as it is to the California isolated virus that they report it as being most closely related to.

Lines 254 to 257: The sentence “The 255 two most phylogenetically related NA genes were from influenza viruses isolated from Northern Shovelers (Spatula clypeata) in California (99.04%) and Illinois (98.76%) (MK995843.1)(Fig 3).” Does not agree with the data presented in their figure. In their figure the virus isolated from Northern Shovelers (Spatula clypeata) in California (99.04%) is the most closely related database sequence used in their analysis. The authors own analysis shows there are seven other database sequences just as related to theirs as the Illinois sequence is. Please reformulate your comments accordingly.

Lines 258 to 261: This sentence as it is written is also miss-leading and inaccurate. I suggest it is corrected to: “Phylogenetic analysis of the A(H2N2) virus obtained from PNLP showed that its HA clustered with multiple virus isolated from with birds sampled in Iceland and North America (Fig 2).”

Lines 267 to 270: the section needs to be revise for accuracy. Deleting the text: “and the PB1 (CY149642.1) 270 and PA (CY149641.1) to and AIV isolated in Canada in 2011 (99.18% 271 and 99.14%)” from the end of the sentence could resolve this issue.

Lines 271 to 274: this sentence also needs to be modified for accuracy I suggest changing it to something like: “The NS (KX620095.1) and NP (KX620073.1) segments were most similar to Brazilian viruses also collected from Lagoa do Peixe National Park in 2012.

Discussion

Line 286: “coast are reservoir” should be changed to: “coast are a reservoir”.

Line 292: The sentence “Of note, semipalmated sandpipers also breed 293 in the southern tundra in Canada and Alaska, and winter in 294 coastal South America” seems out of place and should be deleted.

Line 325: This sentence needs to be reformulated for clarity: “There are still many gaps to be filled in our understanding of viral gene flow between the two hemispheres, but demonstrates the importance of the Atlantic route as a corridor for the movement of AIVs between North America, South America and even Europe.” I am not entirely sure what is being said here, but perhaps the sentence could be revised to something like this: “There are still many gaps to be filled in our understanding of viral gene flow between viruses circulating in the two hemispheres, but our work contributes to understanding of the importance of the Atlantic route as a corridor for the movement of AIVs between North America, South America and even Europe.”

Line 330: I think the closing sentence of the discussion is both redundant and difficult to understand. I think it should be reformulated or deleted.

Figures

Most of the figures seemed to be cropped. While this way of presenting phylogenetic results is undesirable generally, in most cases I do not believe it prevents the authors from showing their key results (i.e the phylogenetic placement of their recovered virus sequences) in this paper. It is, however, a problem for figure S4 and figure S6 where the phylogenetic placement (in a bootstrap-sported clade) of the one of the two study viruses can not be seen. Please revise all the figure captions to clarify how the images have been cropped and revise figures S4 and S6 so that PNLP isolated virus sequence in S4 can be seen in a bootstrap supported clade (and all sequences of this clade can be seen). Please also revise figure S6 similarly so that the PNRJ sequence can be seen in a bootstrap supported clade (and all sequences belonging to this clade can be seen). Also please use consistent labelling of the Matrix segment. In the text and in figure S4 it appears as only “M” but in the legend it is referred to as Matrix, without the abbreviation being highlighted. It would also be helpful in the full name of the genes were provided in all the figure captions.

7. PLOS authors have the option to publish the peer review history of their article (what does this mean?). If published, this will include your full peer review and any attached files.

Reviewer #3: No

---

## [Author Response · Author response to Decision Letter 1]

23 Feb 2024

Response to Reviewers

Reviewer #3: I believe the manuscript is, in general, well-organized and written in good clear easy-to-follow English and I believe the data contained in the manuscript is valuable and will be considered a welcome contribution to the field. I believe the analysis used was appropriate for the study and that the key conclusions the authors have drawn are supported by the authors data and reasonable. However, many of figures are inadequately presented and some of the interpretation of the phylogenetic analysis is incorrect. Labelling of the virus sequences and referencing of viruses is also inconsistent which makes the article unnecessarily difficult to understand in places. Below I am providing detailed comments about how I think the authors can improve their manuscript and indeed resolve their phylogenetic interpretation issues so that the authors can produce a revised manuscript that I would be delighted to recommend for publication in PLoS One.

Answer: Thank you for your valuable feedback. We appreciate your insights and will carefully address the concerns you raised regarding the figures and interpretation of the phylogenetic analysis. We thoroughly reviewed and made the necessary improvements to ensure the accuracy and clarity of our presentation. Your input is crucial, and we are committed to delivering a revised and improved version of the manuscript.

Reviewer #3: Abstract

Overall, I feel there is insufficient information about what exactly has been found. For example, there is no mention of the fact genomic sequences for these viruses have been determined and no direct mention of the results obtained from the author´s phylogenetic analysis (only a mention of the conclusions drawn from this analysis). As PLoS One allows 300-word abstracts and the current abstract is only 169 words so there is plenty of space to provide this detail. I recommend, thus, that they expanded their existing abstract to include additional details.

Answer: We agree with the suggestions in the abstract. We expanded additional details and included new information about our findings. The abstract now has 275-word.

Reviewer #3: Line 37: First sentence. The word “potential is redundant and should be deleted.

Answer: The word “potential” was removed as suggested.

Reviewer #3: Introduction:

Line 84: Although I understand what the authors intend to say here, I think clarifications are needed. I suggest the sentence “Unlike most AIV subtypes, however, North American sequences can be found in both classic North America and Eurasian clades as a result of a trans-hemispheric transmission event and subsequent proliferation” should be changed to: “Unlike most AIV subtypes, however, “North American” clade sequences can be found in both classic North America and Eurasian viral genomes as a result of a trans-hemispheric transmission event and subsequent proliferation”.

Answer: Thanks for this point. As a suggestion, we added this sentence in our text “Unlike most AIV subtypes, however, “North American” clade sequences can be found in both classic North America and Eurasian viral genomes as a result of a trans-hemispheric transmission event and subsequent proliferation”.

Reviewer #3: Line 99: Breeds should be corrected to “breed” and “migrates” to “migrate”.

Answer: Done.

Reviewer #3: Line 104: The last two sentences of the penultimate paragraph of the introduction need to be reformulated for English language clarity: “They arrive on the southern coast of Brazil between November and January where keep together with several species such as gulls, terns, and shorebirds with directly contact, refuel, and then continue south to Patagonia. Their extensive migratory routes and susceptibility to 108 viruses emphasize their role as transcontinental vectors of avian viruses.”

Answer: The sentence was reformulated for the English language to clarify and better understand “They arrive on the southern coast of Brazil between November and January, where they congregate with various species such as gulls, terns, and shorebirds. During this time, they engage in direct contact, refuel, and subsequently continue their migration southward to Patagonia. Given their extensive migratory routes and susceptibility to viruses, these birds play a significant role as transcontinental vectors of avian viruses (see figure 1)”

Reviewer #3: Lines 111-115. The first three sentences of the last paragraph of the introduction need to be revised for clarity. I suggest changing it to something like this: In this study, two A(H2) viruses were isolated from wild-caught white-rumped sandpipers (Calidris fuscicollis). One was an A(H2N1) virus obtained from a bird caught in Restinga de Jurubatiba National Park (PNRJ, 113 Rio de Janeiro); the other, an A(H2N2) virus, was isolated from a bird caught in Lagoa do Peixe National Park (PNLP, Rio Grande do Sul) in the extreme south of Brazil.

Answer: The sentence was substituted as a suggestion in the new version “…In this study, two A(H2) viruses were isolated from wild-caught white-rumped sandpipers (Calidris fuscicollis). One was an A(H2N1) virus obtained from a bird caught in Restinga de Jurubatiba National Park (PNRJ, Rio de Janeiro); the other, an A(H2N2) virus, was isolated from a bird caught in Lagoa do Peixe National Park (PNLP, Rio Grande do Sul) in the extreme south of Brazil…”.

Reviewer #3: Fig 1. There seems to be no reference to the bird migration routes shown in figure 1. I suggest adding a “(see figure 1)” to the end of the last sentence of the penultimate paragraph and providing some description of these routes in the figure caption. Alternatively, the authors could revise the figure (removing this information) and provide more information on the collection site, if this is the only purpose the figure is to serve.

Answer: Thanks for this point. As a suggestion, we added “(see Figure 1)” in this last sentence, and we have described the figure caption.

Reviewer #3: Methods

Phylogenetic analysis section 

Line 19: It appears that the authors used nucleotide sequence alignments (not inferred protein sequences) for the phylogenic analysis; however, they don´t make this explicit. They also state they used 8 AIV segments but do not provide the sequence alignments that they used for these trees. Including the sequence alignment would allow others to build on their study and so I recommend that they are all included in the supplementary material. Failing this, I think the authors should at least provide information of the length of the sequence alignment used to construct each of their phylogenetic trees.

Answer: We agree on this point. We added new information in the sentence “…Maximum Likelihood analysis was conducted using IQ-TREE [24] for all 8 entire nucleotide AIV segments of each virus…”. In addition, we added the length of the sequence alignment used to construct each of the phylogenetic trees on the caption of each figure. 

Reviewer #3: Line 192: the sentence “Bootstrap was set to 1000 for statistical significance” needs to be revised for accuracy/clarity. I suggest changing to something like: “The statistical significance of phylogenetic groupings was tested with bootstrap analysis using 1000 replicates”.

Answer: The sentence was changed to “The statistical significance of phylogenetic groupings was tested with bootstrap analysis using 1000 replicates” as suggested.

Reviewer #3: Results

Sampling

Line 218: Presumably all samples were tested with the RT-PCR described in the methods section; however, I feel I would be helpful to make this explicit here in the results section.

Answer: Yes, we agree. The new sentence about RT-PCR screening was added “…All samples were tested for avian influenza virus RT-PCR. Two samples, one from PNLP and one from RJNP, were positive for the H2 subtype, both from free-ranging white-rumped sandpipers (C. fuscicollis)…”.

Reviewer #3: Line 219: The reference to figure 1 at this juncture does not make sense, please delete.

Answer: Done. Was deleted.

Reviewer #3: Phylogenetic analysis section

Line 223: Presently reads thus: “Phylogenetic analysis of the A(H2N1) virus showed that its internal genes clustered not with those from viruses in South America, but with those of influenza viruses isolated from North America (Fig 2 and 3)”. I think this should be revised for clarity to something like: “Phylogenetic analysis of the A(H2N1) virus recovered from the PNRJ capture bird showed that all of its internal genes clustered not with gene sequence isolated from birds sampled in South America, but with sequences of influenza viruses isolated from birds sampled in North America (Fig 2, 3, SF1-5).

Answer: The sentence was reformulated as suggestion “Phylogenetic analysis of the A(H2N1) virus recovered from the PNRJ capture bird showed that all of its internal genes clustered not with gene sequences isolated from birds sampled in South America, but with sequences of influenza viruses isolated from birds sampled in North America (Fig 2, 3, SF1-5).”

Reviewer #3: Lines 241 to 250: The information provided here is not phylogenetic analysis or helpful for interpreting the authors data. I strongly recommend that it is deleted as some of it is, in fact, miss-leading. For example, a correct interpretation of the phylogenetic analysis shown in supplementary figure 5 is that the PA gene of A(H2N1) virus from PNRJ is equally closely related to a Mississippi isolated virus as it is to the California isolated virus that they report it as being most closely related to.

Answer: We thank the reviewer for the recommendations, which were considered. Initially, we used two strategies to characterize these samples: (i) phylogenetic analysis; and (ii) the identity at the nucleotide level observed between the sequences considered (in percentage). Based only on identity, it is possible to observe that the PNRJ sequence presents a greater identity value (99.33%) with the California isolate (MK995846.1) in 2017 than the PA gene (MN431162.1) in 2018 (99,02%) isolated in Mississippi. However, phylogenetic analysis indicates that the PNRJ sequence is equally related to both the Mississippi isolate and the California isolate, thus providing a clearer scenario through this inference for the evolutionary pattern observed for this highlighted clade. The phylogenetic tree was indicated in the figures on the left, and on the right, the graphic representation is a zoom-in of the larger tree, which allows us to observe the phylogenetic relationship of our sequences to other sequences previously sequenced and deposited in banks of public data. In addition, we included “Identity and Phylogenetic analysis” in the title session to better understand.

Reviewer #3: Lines 254 to 257: The sentence “The 255 two most phylogenetically related NA genes were from influenza viruses isolated from Northern Shovelers (Spatula clypeata) in California (99.04%) and Illinois (98.76%) (MK995843.1) (Fig 3).” Does not agree with the data presented in their figure. In their figure the virus isolated from Northern Shovelers (Spatula clypeata) in California (99.04%) is the most closely related database sequence used in their analysis. The authors own analysis shows there are seven other database sequences just as related to theirs as the Illinois sequence is. Please reformulate your comments accordingly.

Answer: Thanks for this point. The sentence was corrected to clear and removed “…Illinois (98.76%)…”. The new version represent “…The most similarity related NA gene was from influenza viruses isolated from Northern Shovelers (Spatula clypeata) in California (99.04%) (MK995843.1) (Fig 3).…”. 

All seven sequences within the clade showed a similarity below 98.76%, as found on the Illinois sequence (MG280504). However, we realized that this statement could lead to ambiguity due to the tree structure and removed it from the text.

Reviewer #3: Lines 258 to 261: This sentence as it is written is also miss-leading and inaccurate. I suggest it is corrected to: “Phylogenetic analysis of the A(H2N2) virus obtained from PNLP showed that its HA clustered with multiple virus isolated from with birds sampled in Iceland and North America (Fig 2).”

Answer: Thanks for suggestion. The sentence was reformulated in the new version “…Phylogenetic analysis of the A(H2N2) virus obtained from PNLP showed that its HA clustered with multiple viruses isolated from birds sampled in Iceland and North America (Fig 2). Based on the identity at the nucleotide level analysis, the A(H2N2) virus showed its HA most closely with that A/lesser black-backed gull/Iceland/1597/2012 (H2N7) with 98.51% identity (CY149380.1)…”

Reviewer #3: Lines 267 to 270: the section needs to be revise for accuracy. Deleting the text: “and the PB1 (CY149642.1) 270 and PA (CY149641.1) to and AIV isolated in Canada in 2011 (99.18% 271 and 99.14%)” from the end of the sentence could resolve this issue.

Answer: The sentence was deleted as suggested. The new version represent “…Other gene segments of the A(H2N2) virus were most similar to viruses detected in North America; for example, the PB2 segment was most closely related to an AIV isolated in Mississippi (CY133700.1) in 2011 (98.83% identity) (supplemental material)….”

Reviewer #3: Lines 271 to 274: this sentence also needs to be modified for accuracy I suggest changing it to something like: “The NS (KX620095.1) and NP (KX620073.1) segments were most similar to Brazilian viruses also collected from Lagoa do Peixe National Park in 2012.

Answer: The sentence was modified to “…The NS (KX620095.1) and NP (KX620073.1) segments were most similar to Brazilian viruses also collected from Lagoa do Peixe National Park in 2012…”

Reviewer #3: Discussion

Line 286: “coast are reservoir” should be changed to: “coast are a reservoir”.

Answer: Done.

Reviewer #3: Line 292: The sentence “Of note, semipalmated sandpipers also breed 293 in the southern tundra in Canada and Alaska, and winter in 294 coastal South America” seems out of place and should be deleted.

Answer: The sentence was removed.

Reviewer #3: Line 325: This sentence needs to be reformulated for clarity: “There are still many gaps to be filled in our understanding of viral gene flow between the two hemispheres, but demonstrates the importance of the Atlantic route as a corridor for the movement of AIVs between North America, South America and even Europe.” I am not entirely sure what is being said here, but perhaps the sentence could be revised to something like this: “There are still many gaps to be filled in our understanding of viral gene flow between viruses circulating in the two hemispheres, but our work contributes to understanding of the importance of the Atlantic route as a corridor for the movement of AIVs between North America, South America and even Europe.”

Answer: The sentence was changed as suggested.

Reviewer #3: Line 330: I think the closing sentence of the discussion is both redundant and difficult to understand. I think it should be reformulated or deleted.

Answer: The sentence was removed.

Reviewer #3: Figures

Most of the figures seemed to be cropped. While this way of presenting phylogenetic results is undesirable generally, in most cases I do not believe it prevents the authors from showing their key results (i.e the phylogenetic placement of their recovered virus sequences) in this paper. It is, however, a problem for figure S4 and figure S6 where the phylogenetic placement (in a bootstrap-sported clade) of the one of the two study viruses can not be seen. Please revise all the figure captions to clarify how the images have been cropped and revise figures S4 and S6 so that PNLP isolated virus sequence in S4 can be seen in a bootstrap supported clade (and all sequences of this clade can be seen). Please also revise figure S6 similarly so that the PNRJ sequence can be seen in a bootstrap supported clade (and all sequences belonging to this clade can be seen). Also please use consistent labelling of the Matrix segment. In the text and in figure S4 it appears as only “M” but in the legend it is referred to as Matrix, without the abbreviation being highlighted. It would also be helpful in the full name of the genes were provided in all the figure captions.

Answer: Thanks for the suggestions. All figures and captions have been revised and greater attention was paid to figures S4 and S6 as suggested by the reviewer. These two have been re-formatted and improved for better understanding. The bootstraps are included in new version.

---

## [Editor Report · Decision Letter 2]

5 Mar 2024

PONE-D-23-19373R2Evidence of reassortment of avian influenza A (H2) viruses in Brazilian shorebirdsPLOS ONE

Dear Dr. de Araujo,

Thank you for submitting your manuscript to PLOS ONE. After careful consideration, we feel that it has merit but does not fully meet PLOS ONE’s publication criteria as it currently stands. Therefore, we invite you to submit a revised version of the manuscript that addresses the points raised during the review process.

We look forward to receiving your revised manuscript.

Kind regards,

James Lee Crainey, Ph.D.

Academic Editor

PLOS ONE

Journal Requirements:

**Additional Editor Comments:**

While I really do not want to delay the publication of this manuscript any further and I feel that almost all of the concerns raised by the referees have been addressed, I am afraid the manuscript still does not meet the PLOS One criteria for publication. The paper´s interpretation of some the presented phylogenetic analysis is still inaccurate. The quickest way to resolve this would be to return the newly titled “Identity and phylogenetic analysis” section to its original form i.e “Phylogenetic analysis” and to delete two sections of text:

Detele this passage “The PB2 gene was most closely related to an AIV isolated in Wisconsin (MT824582.1) in 2018 (99.57% identity), the PB1 to an AIV isolated in Ohio (MN430996.1) in 2018 (98.28%), the PA to the NP to an AIV isolated in Wisconsin (MK237257.1) in 2017 (97.18%), the M to an AIV isolated in North Dakota (MN253700.1) in 2015 (98.73%) and the NS to an AIV isolated in Minnesota (MN254207.1) in 2015 (98.53%) (S1- S6 Figs). The closest database entry to the HA gene came from A/blue-winged teal/Guatemala/CIP049-H121-36/2014 (H2N9), but with only 93.34% identity (MK326705.1) (**Fig 2**).”Delete this text: “for example, the PB2 segment was most closely related to an AIV isolated in Mississippi (CY133700.1) in 2011 (98.83% identity) (supplemental material)”

If these minor changes were made, I would be happy to accept the manuscript. Alternatively, the two highlighted sections of text could be revised. If this option is chosen, the revised passages would need to be revised so that they are no longer in conflict with what is shown in the supporting figures. It is fine to say that a gene shares X% identify with another gene, but not to ok to say it is most closely related to it if that conflicts with supporting phylogenetic analysis. Also, if the methods section sub-title “identity and phylogenetic analysis” is to be preserved, I think there should be some description of how any identity analysis was performed.

---

## [Author Response · Author response to Decision Letter 2]

5 Mar 2024

To Editor:

PLOS ONE

Dear James Lee Crainey, Ph.D.

Academic Editor

Thank you for your thorough review and valuable feedback on our manuscript. We sincerely appreciate your efforts to ensure the quality and accuracy of published research in PLOS One. We are re-submitting the final revised manuscript entitled “Evidence of reassortment of avian influenza A (H2) viruses in Brazilian shorebirds” by Thomazelli et al., 2023, for consideration as a research article in PLoS ONE. 

While we share your eagerness to move forward with the publication process, we understand the importance of addressing any lingering concerns regarding the accuracy of our phylogenetic analysis. We have carefully considered your suggestions and agree that revisions are necessary to meet the PLOS One criteria for publication. To address the concerns raised, we propose the following revisions:

We will revert the section titled "Identity and phylogenetic analysis" back to its original form, "Phylogenetic analysis," to maintain clarity and accuracy in our presentation. We removed the specified passages regarding the relationships "between the PB2 gene and the AIV isolated in Wisconsin, as well as the PB2 segment and the AIV isolated in Mississippi." We understand the importance of ensuring our statements align with the supporting phylogenetic analysis. Once these revisions are made, we resubmit the manuscript for your review. We are confident that these adjustments will address the concerns raised and bring the manuscript in line with the standards of PLOS One.

Thank you once again for your invaluable feedback and guidance throughout this process. We look forward to your positive answer input and the opportunity to share our research with the scientific community.

Best regards,

Jansen

Prof. Jansen de Araujo

Laboratório de Pesquisa em Vírus Emergentes- LPVE

Institute of Biomedical Science

University of São Paulo State, Brazil

Prof. Lineu Prestes avenue, 1374, São Paulo, USP- Brasil

e-mail: jansentequila@usp.br

---

## [Editor Report · Decision Letter 3]

7 Mar 2024

Evidence of reassortment of avian influenza A (H2) viruses in Brazilian shorebirds

PONE-D-23-19373R3

Dear Dr. Araujo

We’re pleased to inform you that your manuscript has been judged scientifically suitable for publication and will be formally accepted for publication once it meets all outstanding technical requirements.

Kind regards,

James Lee Crainey, Ph.D.

Academic Editor

PLOS ONE

Additional Editor Comments (optional):

I have reviewed your latest version of your manuscript and am now satisfied that it addresses all of the reviewers concerns and that it meets PLOS One´s criteria for publication. I therefore only wish to thankyou for your patience and congratulate you all on your fine contribution to the field.

My reading of your cover letter makes  me think that the fact you have not changed the methods section subtitle “identity to phylogenetics” back to “phylogenetics” is an oversight that you will correct at the proof stage. Please feel free to make this change at the proof stage if you wish to.
---

## [Editor Report · Acceptance letter]

21 Mar 2024

PONE-D-23-19373R3 

PLOS ONE

Dear Dr. de Araujo, 

I'm pleased to inform you that your manuscript has been deemed suitable for publication in PLOS ONE. Congratulations! Your manuscript is now being handed over to our production team.

Kind regards, 

on behalf of

Dr. James Lee Crainey 

Academic Editor

PLOS ONE